# Innovation of the Third Sector to Improve Nature Reserve Management in the Post-COVID-19 Epidemic Era

**DOI:** 10.3390/ijerph192316278

**Published:** 2022-12-05

**Authors:** Tian Guo, Zhitao Zheng, Yourui Zheng

**Affiliations:** School of Law, Hainan University, Haikou 570228, China

**Keywords:** post-COVID-19 epidemic era, sustainable development, social enterprises, protected areas, public interest

## Abstract

Encouraging the strong recovery of the economy is an urgent priority for all nations in the post-COVID-19 epidemic era. Social enterprise, as a new-type third sector, boasts unique advantages in structure and function, which can reach public interest targets without relying on government spending; social enterprises can effectively reduce the cost of ecological construction and maintenance, provide more professional and diverse services, and promote sustainable development in the regional economy, society, and ecology. Through an analysis of their structure and function, this study proved that social enterprises serve as a significant institutional innovation to cure the “government failure” and “market failure” in the area of public interest with the merits of the simple structure of the main body, strong self-innovation ability, high spontaneity of members, and convenient application of laws. Thus, its introduction to nature reserve management can pragmatically relieve the financial pressure and increasingly achieve public interest goals. The empirical research indicates that social enterprises need to be supplemented with the maintenance mechanism of bidirectional targets so that they can do their best to meet the requirements of “low government spending, high ecological benefits” in constructing nature reserves in the post-COVID-19 epidemic era, fully motivate the market, and develop the reliable force for public welfare.

## 1. Introduction

The COVID-19 outbreak has lasted for more than two years, posing a grave threat to human life and health and crippling the world economy. The epidemic caused a collapse of the amount of global trade and investment, as well as a decline in consumption, with social problems, such as fairness, distribution, efficiency, employment, and environmental management becoming more prominent simultaneously. Therefore, people long for the advent of the post-COVID-19 epidemic era, where the COVID-19 pandemic has been basically controlled, the virus mutation can be addressed by the social medical conditions, the global economy and trade exchanges can resume normally, and people can create a new way to live and produce [1]. In the post-COVID-19 epidemic era, where COVID-19 fluctuates unceasingly, we can also maintain the vitality of the market economy, the momentum of social development, and a healthy ecosystem, that is, following the sustainable development goals. Therefore, promoting a strong economic recovery is a priority as well as a major goal of all governments in the post-COVID-19 epidemic era.

On 1 June 2022, China’s National Forestry and Grassland Administration issued the Interim Measures for the Administration of National Park (hereinafter referred to as the “Measures”). This administrative regulation indicates that China’s work on the establishment of a system of protected areas represented by national parks has developed from top-level design to a deeper level. Based on the basic principle of “multi-party involvement” proposed by the Measures, the regulation meets the basic demand that the organizations in reserves should insist on “government-leading, multi-party participation”, which is raised in the Guiding Opinions on the Establishment of a Nature Reserve System with National Parks as the Main Body (hereinafter referred to as the “Guiding Opinions”) [2] issued by the government. The important reason lies in the fact that the construction of protected areas, as a high-cost public interest project, accomplishes this now and will benefit our future generations. It will be under great pressure if it only relies on government fiscal input in the post-COVID-19 epidemic era when the government struggles with various difficulties in economic resumption. Furthermore, the conservation model of “the government direct control” can help form a consistent responsibility system to finish the unified conservation goal, which infuses the governance with more stability and authority [3], but this system may ignore public opinions and risk regulation failure and rights abuses. For that reason, it is necessary for social entities to actively join in the conservation, maintenance, and particularly daily operation of the protected areas, and the green channel for the law and policy has been opened.

Multi-party participation can stimulate people’s behavioral initiative [4], responsibility, and creativity at the psychological level, and also avoid aggravating management pressures with the expansion of government agencies but keeping social stability through mutual restraint and supervision [5]. Furthermore, thanks to the multi-party participation, it manages to combine the interests of different levels with the rules to make key decisions that can avoid decision failure in terms of the established institutions and inner boundaries. The state-led system and the multi-party participation can be seen as the relationship between the “body” and “wing”, that is, the need for inextricable cooperation of them in the composition of the institution of protected areas. The new approach, therefore, which lets the governments play a guiding role in the construction, management, supervision, conservation, and investment solicitation while the social institutions develop diverse ways of “public interest governance, community governance, and joint governance”, will be the mainstream in the protected areas’ construction in the post-COVID-19 epidemic era.

## 2. Literature Review and Theoretical Analysis

### 2.1. Background of the Social Enterprise

In the 1970s, European and American countries were mired in the “welfare states’ crisis”. Before that, European and American countries followed the traditional public governance model and established the omnipotent government (also known as the “big government doctrine”), which provided all public services [6]. However, economic stagflation reduced national revenues so much so that the government failed to maintain higher input costs in infrastructure and social services; the intended political intervention to correct the deficiencies of the market mechanism in the distribution field did not work anymore in the crisis, and the government lost the flexibility to respond to social problems, with the advent of government failure [7]. Then, the new public management theory claimed that the adoption of the market mechanism in the public area would reduce the government’s financial burden and solve economic stagnation by outsourcing public services [8]. However, it is difficult for enterprises pursuing profit maximization under the market mechanism to adhere to the principle of giving priority to the public interest. Unfair competition, such as an oligopoly or even monopoly, makes public services overpriced and the distribution unfair, which leads to new problems, such as market failure [9]. Against that background, the left-wing parties in Europe and the Democratic Party of the United States proposed a “third way” to overcome the crisis, that is, to pursue multi-party cooperation across classes, strike a balance between the role of the government and the role of the market, the state-owned economy, and the private economy, provide positive welfare through extensive participation in the process of economic development, and enhance social fairness [10]. Thus, the new public service theory responded to the call of the time, namely that the government should work with the third sector to deliver public services through democratic participation or collaboration [11]. However, at this time, the supply chains of practice entities could not meet the demand. Non-profit organizations relied heavily on government finance and lacked self-innovation ability. After the market financing, non-profit organizations lost their autonomy due to the diversification of entities and fell into the predicament of mission drift and voluntary failure [12] (Table 1).

Thanks to continuous experiments by the government, market, and social sectors, the social innovation spirit emerged under the combination of “independence”, the “mutual benefit” spirit of commercial organizations, and the “voluntary” “public interest” purpose of non-governmental organizations, and then further developed a hybrid organizational model [13]. Developed countries are moving towards a new direction of “entrepreneurial government”, in which the government does not directly provide public services, but only provides resources and policy tools, the demand and supply of public interest are regulated by market mechanisms, and the stability of public–private cooperation is maintained by a legal framework. Non-profit organizations need to seek self-development in the market mechanism and assist enterprises to balance social interests and commercial profits. Hence, the cooperation of the three makes public services realize low cost, high efficiency, and fair and sufficient supply [14]. The biggest difference is that in the new public management or new public service theory, for-profit enterprises and non-profit organizations are regarded as two departments operating independently. However, the entrepreneurial government promotes the mutual learning and integration of for-profit and non-profit organizations, breaking through the traditional division of organizations to generate social enterprises [15].

### 2.2. Concept and International Practice of Social Enterprises

Social enterprise research can be traced back to the 1980s, but it was introduced to China in early 2000. Now there is no common definition of social enterprises, but professional associations propose the basic characteristics of the ideal social enterprises with the combination of their birth and practice development. The Department of Trade and Industry (DTI) believes that social enterprises serve as commercial organizations aimed at the public interests [16]. Focusing on profit-making, there are three authoritative interpretations of the concept and criterion of social enterprises. Firstly, according to the Organization for Economic Co-operation and Development (OECD), social enterprises generally refer to all social and economic organizations whose primary purpose is not to maximize profits but to achieve certain social and economic goals and bring about innovative solutions [17]. The concept of social enterprises from the OECD shows the most broadness, as it takes the enterprise strategy, a subjective factor, as the only criterion. Since the operation of the OECD began 30 years ago, there have been more than 70,000 certified social enterprises in the UK and more than 50,000 in Spain [18]. Secondly, Professor Ze Lai, the pioneer researcher of social enterprises, said that social enterprises act as organizations that reach social objectives through commercial means [19]. Based on the concepts proposed before, some scholars have suggested that whether an organization is a for-profit entity is not the criterion, but rather whether an organization uses commercial means to achieve social objectives and maintains its operation with commercial profits [20]. Thus, the related academics pay more attention to the objective criterion of “commercial means” [21]. Thirdly, Professor Divney, the research director of the targeted socio-economic research (TSER) program in the European Commission, said that social enterprises need to focus on both the entrepreneurial side and social indicators. The former includes the following: (1) continuous activities of goods production or service sales; (2) high autonomy; (3) significant economic risks; (4) minimization of the proportion of paid employees. The latter includes the following: (1) clear objectives that benefit the community; (2) innovative solutions initiated by groups; (3) decision-making rights which are not subject to the capital equity but adopt the one man, one vote rule; (4) having the “participation” of those who are affected by the enterprise activities; (5) limited profit distribution, which is the most stringent criterion. China began to study social enterprises around the year 2000, when professors Jitong Liu and Lirong Shi undertook the initial exploration. Basically speaking, China followed the concept of social enterprises abroad to form two criteria. Firstly, the main purpose of the organizations is to achieve a social interest. Secondly, the organizations capitalize on commercial operations and then invest the proceeds into specific goals of public interest. Through the characteristics of “targets + methods”, the China Charity Fair (CCF) defines social enterprise as “an enterprise or social organization that takes solving social problems as its primary goal in the innovative solution that conforms to the entrepreneurial spirit” [22]. The definition of social enterprise is shared by scholars and research institutions at home and abroad, and this article believes that social enterprise can be given as a concept with outstanding core features and wide coverage, that is a “social enterprise is a commercial organization, which uses commercial means to obtain income and profit, and its revenue is not only used in the organization’s operation and expansion, but more significantly, its primary goal is to achieve social welfare purposes”.

In practice, the main types of social enterprises in Europe include associations, cooperatives, community interest companies (CICs), etc. [23]. In 1991, Italy became the first country to enact the relevant law and set up “social cooperatives”; in 1995, Belgium introduced the concept of the “social purpose company”, and Portugal added “social solidarity cooperatives”. In 1999, Greece added “social cooperatives with limited liability” [21]. Britain has the largest scale of social enterprises. In 2001, the British parliament formulated a development strategy for social enterprises, proposing a definition of social enterprises by the Department of Trade and Industry, and giving it a legal form, the “community interest company”. Then, these measures were taken by other countries. The concept of social enterprise was introduced in the United States in the 1990s, which attracted great interest from the academic world and public interest organizations. In 1993, Harvard Business School put forward the “Social Enterprise Program”, and Stanford, Yale, and other prestigious universities have successfully set up related research and training programs, which have won strong support from many foundations [24]. From 2008 onwards, social enterprise forms, such as low-profit limited liability companies (L3C) and benefit corporations began to appear in state legislation. Since 2008, social enterprise forms, such as low-profit limited liability companies (L3C) and benefit corporations began to appear in state legislation. Social enterprise law in the United States has not been upgraded to federal law, and there are no hard provisions on making profit. Therefore, the forms of social enterprises are more flexible and diverse; general for-profit companies that undertake social responsibilities, such as donating or providing community services, companies that have both for-profit and social goals (standard), and non-profit organizations that use commercial means to support public interest goals, such as foundations that use stock financing, can all be recognized as social enterprises and enjoy preferential policies.

Social enterprises in China started late, but have developed quickly. In 2015, the Fourth Charity Fair (hereinafter referred to as the “CCF”) initiated the China Charity Fair Social Enterprise Certification and issued the Measures for China Charity Fair Social Enterprise Certification [22]. These actions forge a comprehensive platform for social enterprises in the audit, connection, and development, and provide a cushion for certified enterprises in funds, personnel, and incubation. The certification manual issued by the government in 2018 proposed the standards for social enterprises (standard version 1.0), as follows. Firstly, social enterprises must adopt a registration system, and the governance structure should be under supervision with taxes and social security. Specifically, social enterprises should be registered and established in China (including Hong Kong, Macao, and Taiwan) for at least one year, with a full-time salaried team of at least three people and a sound financial system, as well as the independent accounting. Secondly, social enterprise should be organized for public interest purposes and be equipped with management mechanisms to ensure its robustness. Specifically, social enterprises should pay attention to the areas suffering from both government failure and market failure, and actively change the external environment to solve social problems. Social enterprises need to establish the mechanisms with diversity and transparency in order to accept social supervision. Furthermore, the profits related to the social goals should implement limited distribution, and the related assets should also be locked. Thirdly, social enterprises should provide a specific product or service by making a profit. The certification standard of the CCF focuses on enterprise innovation, which requires social enterprises to cooperate among different fields and to innovate technology, process, and mode. The financial sustainability of the enterprises should be judged by the growth in revenue and the number of employees. Fourthly, social enterprises should have a positive impact on society. The measures require social enterprises to have a clear output and social influence, with clear and measurable achievements. To facilitate the identification of customers and the general public, the CCF makes a quantitative assessment on social enterprises every year and awards the certificates and a special logo. By December 2018, the CCF had certified more than 1400 enterprises and institutions, of which 238 had passed, covering 71 cities in 26 provinces. It covers 14 areas of public welfare, including eco-development, disability services, community development, commonwealth finance, old-age care, youth education, assistance to disadvantaged groups, rural development, the Internet, arts and culture, and women, children, and families (Figure 1). In 2018, the CCF was officially recognized as an evaluation unit for social enterprises in Beijing [25] and took charge of the evaluation and accreditation of social enterprises in Chengdu, the capital city of Sichuan Province [26]. Since 2019, the China Social Enterprise Certification Center (CSECC), the first national private certification system in China, has taken over the CFF certification work. In 2020, it combined with the standards of local certification bodies and other professional institutions to put forward the four-dimensional certification standard (standard version 2.0), including the goals for environment and sustainable development, value creation and profit distribution, social missions, and social enterprise shareholders (Figure 2). The fresh certification standard not only makes comprehensive progress in the indicators of version 1.0 but also indicates the in-depth development of social enterprises in China, with the deep integration of industry standards and local standards.

To sum up, although the development and business scope of social enterprises vary from country to country, the common understanding of social enterprises is as follows: a non-profit, private organization that directly provides, in the form of a business, goods or services closely related to its public interest purposes, such as elderly care, book sales, or training courses. Investors and other interest entities can join the governance structure and bear the economic risks brought by enterprise operations.

## 3. Advantages of Social Enterprises—Through Comparison with Foundation Model in Practice

The particularity of social enterprises is that they use commercial means to achieve public interest purposes. Compared with traditional public interest organizations, they are more suitable to participate in the operation of national parks. In the UN’s Sustainable Development Goals, social enterprises need to combine ecological conservation with economic construction to achieve a harmonious existence and healthy interactions between man and nature in the process of transforming natural resources into economic benefits [27]. Social enterprises contribute to the coordinated operation of the “ecological-economic-social” three-dimensional composite system [28], to achieve the targets of the nature reserve system reform. Taking the management practices of Laohegou Nature Reserve as an example, this paper proves the advantages of social enterprises through comparative methods. Laohegou Nature Reserve is the first nature reserve in China to be led by a foundation, with an obvious “private” nature, which shows great characteristics in its management structure, community participation mode, and capital operation form. However, there are some problems, such as an imbalance of income and expenditure, unbalanced community development, and a lack of citizen participation. The foundation as the leading organization is limited by its own functions, and the overlapping of laws may set obstacles to the innovation of business models.

### 3.1. Development and Transformation of Laohegou Nature Reserve

Laohegou Nature Reserve is located in Pingwu County, Mianyang City, Sichuan Province, covering an area of about 153 square kilometers. The Motianling Mountains, where the reserve is located, boast a vertical height difference of about 2500 m [29]. The rich climatic conditions have produced vertical distribution zones of forests with high biodiversity. In general, there were three development stages of Laohegou Nature Reserve (Table 2). The first stage was the forest farm period, when environment conservation was not the focus of its management and operation. The collective-owned forests scattered in Laohegou, Shanhe Creek, and Xiaohezi Creek constituted the state-owned forests together, as managed by the forestry bureau. The county government was encouraged to cut the natural forests and plant economic forests. The second stage was the stagnation period. When a natural forest conservation program was rolled out nationwide in 1998, logging was prohibited in the area, making it necessary to seek a balance between conservation and livelihood development. Consequently, the local forest-oriented economy suffered badly. At that time, China were experiencing serious pollution and ecological degradation. The primary value orientation of the state-owned forest farm began to diverge from economic development and towards the protection of ecology and environment. It is safe to say that the economic development had made comprehensive concessions for environment conservation at that time. Laohegou also began to focus on managing its man-made forests. The third stage was the nature reserve construction period. In June 2008, the State Council issued the Opinions on Comprehensively Promoting the Reform of the Collective Forest Rights System, which ensured the contracted farmers’ management right and the ownership of forests, and allowed various means used in the forest land transfer, such as subcontracting, leasing, shareholding, and mortgaging. The management of commercial forests and non-commercial forests were classified, and the property rights of collective-owned forests and state-owned forests were separated in Laohegou. The “Laohegou Nature Reserve” was established officially and was supported by government finance. The nature reserve carries out ecological protection and sightseeing activities. It mainly focuses on ecological protection while pursuing common development in both the community and economy.

After 2008, Laohegou embraced its golden age of development. The private capital was first introduced in the operation of Laohegou Nature Reserve. The Nature Conservancy (hereinafter referred to as the “TNC”), the largest global non-profit environmental organization, was invited by the Chinese government in 1998 to provide scientific methods and solutions for ecological and environmental protection [30]. In 2009, with the guidance of the National Forestry and Grassland Administration, Sichuan Forestry and Grassland Administration cooperated with TNC to launch the “Public Benefit Protected Areas Project”, using charitable foundations to establish the nature reserve, creating a development model of “government supervision and private investment management”. However, TNC as a foreign foundation has merely established a representative office in China. According to the Charity Law and the Regulation on Foundation Administration of China, TNC is not allowed to raise or accept donations, and cannot become the management entity of the “Public Benefit Protected Areas”. Given that restriction, TNC cooperated with well-known domestic enterprises, such as Tencent, Ali, Mengniu, and Health Yuan, and established the Sichuan West Nature Conservation Foundation, SWNCF (currently “Sichuan Paradise International Foundation”, hereinafter referred to as the “Paradise”) in September 2011 as the entity for contact and finance. The SWNCF signed an agreement with Pingwu County Government, entered into a 50-year lease of the management rights of state-owned forests and collective-owned non-commercial forests in the nature reserve by means of delegated management, and acquired the use right of fixed assets of the former state-owned forest farm by means of redemption. In 2013, SWNCF managed to make the land a county-level nature reserve. Furthermore, in September 2014, the Paradise established the Laohegou Nature Reserve Center, which became China’s first nature reserve that was established and managed by private, non-profit organizations.

### 3.2. Experience of the Foundation’s Operation Model

As shown in Figure 3, the organizational structure of Laohegou Nature Reserve shows a three-dimensional pattern of “government authorization-foundation management-community cooperation and development” [31]. Through tiered examination and authorization, the forestry department adopts “package” and “contract-issuing” in the management and operation of Laohegou Nature Reserve, and performs the functions of guidance, supervision, and assessment, and also dispatches to the forest police station to carry out grassroots law enforcement. As the contractor, the Paradise Foundation takes the responsibility for the ecological protection of the reserve as well as the improvement of community development. Specifically, it includes financing the working capital of the nature reserve, coordinating the matters of foreign cooperation, and the overall planning of the operation and management of the nature reserve; Laohegou Nature Reserve, as the representative organization, is responsible for the execution of affairs; TNC, the international non-profit organization, is responsible for environment protection behind the Paradise, based on its strong scientific research ability to provide scientific planning and consulting services for Laohegou. The development of the Laohegou Nature Reserve benefits from the effect brought by this three-dimensional structure, and the successful experience can be summarized into the following three items: 

Firstly, the great improvement in management relies on the professional ability of scientific research. For example, relying on TNC’s scientific research strength, the reserve has realized specialization, refinement, and digitalization from site selection, formulation of the conservation plan, and dynamic management of biological resources. Founded in the U.S. through grassroots action in 1951, the TNC has grown to become one of the most effective and wide-reaching environmental organizations in the world. Thanks to more than a million members, staff, and scientists, it impacts conservation in 72 countries in terms of land and water-protected areas, which has produced rich scientific research achievements and practical experience [32]. The first step in the plan is to prioritize the reserve selection. With the Sichuan Biodiversity Conservation Strategy and Action Plan signed by TNC and the Forestry Department of Sichuan Province, it took nearly two years to take the eco-regional assessment (ERA) based on biodiversity, species habitats, environmental threat indices, and socio-economic conditions. Additionally, TNC developed a Conservation Action Plan (CAP) and scientific research reports to serve as a guide for the Laohegou Nature Reserve Center. Then, to attract scientific research institutions to nest and investigate there, TNC and Paradise jointly built a platform where the results from scientific research can be implemented in a timely fashion. Finally, adaptive management has developed in the Laohegou Nature Reserve [33]. Specifically, with the help of the Geographic Information System (GIS), the protected area is divided into grid blocks covering an area of 1 square kilometer, where an infrared trigger camera and automatic acoustic recording device are installed. As such, biological resources can be monitored around the clock and evaluated by indicators with the wildlife AI identification system and patrol software, video surveillance equipment, and the analysis management platform of the data and other tech. With the multiple techniques above, the reserve can keep an eye on the progress of each protection action to improve the plans in a timely fashion.

Secondly, the in-depth exploration of the common development mold in the community occurs. The governments encourage multiple entities to participate in the construction of the protected areas system. One reason is to draw on the knowledge and experience of various entities, and another reason is to explore environmentally friendly economic development methods through them. Using natural resources to transform the environment can cause either exhaustive destruction or a nurturing regeneration of the ecosystem. The ideal development model, therefore, is to combine the benign mutuality between humans and the environment with economic growth on the material plane [34]. For example, by-products, such as featured agricultural by-products and tourism, are substantial ways for local villagers and enterprises to create economic profits and make more people share the development achievements. Hence, whether the relationship with the surrounding communities can be properly handled is a key indicator for evaluating the achievements of social organizations participating in the construction of protected areas. Laohegou Nature Reserve, which focuses on localization and multi-directional financing, plays a demonstration role. First of all, Laohegou Nature Reserve Center as a management agency prefers the former employees of the logging company and people from poor families to be rangers and support staff. It also asks for grassroots protection stations to support the community. Through these efforts, the center reinforces the link with the community. Secondly, in cooperation with Ant Financial Services Group and Alibaba Poverty Alleviation Fund, the Paradise has carried out demonstration projects in poverty alleviation through ecological protection in the Minzhu village nearby, and has established a customized professional cooperative—the Minfu Agricultural Products Cooperative—creating a green industrial chain that “combines production, marketing of featured agricultural products with ecological tourism”. In agreement with the cooperative, Paradise requires that a percentage of its income be allocated to environmental governance. The Laohegou Nature Reserve ties nearby villages into a sustainable economy so that people will not have to venture into the reserve to log. Furthermore, it extends the industrial chain to transform natural resources into quality products.

Thirdly, the reserve speaks actively and influences policy-making. To receive long-term policies from the government is of great importance, as Laohegou Nature Reserve is operated and managed by Paradise. In January 2018, Paradise and the Department of Social Affairs of the National Development and Reform Commission of China signed the Framework Agreement on Developing National Park System Construction Cooperation to conduct national park policy publicity, personnel training, multi-participation policy, and legislation research. Furthermore, Paradise and the government agreed to launch a three-year pilot cooperation in national parks. Furthermore, the Paradise has also enhanced its political influence through its partners. Under the guidance of Tencent, Aliyun, and TNC, the Laohegou Nature Reserve Center uses the Internet and information technology to improve the data processing ability and the management efficiency of the rangers, and to promote intelligent technology products applied in environmental protection work globally, which can demonstrate the achievements of China’s application of the Internet in this field. Nature reserves use Internet firms for scientific and technological management, and Internet firms also hope to help the partners expand their operation items. During the National Congress in March 2018, Huateng Ma, co-chairman, and Guojun Shen, executive chairman, of the Paradise, put forward proposals on the participation of public interest organizations in ecological protection, respectively. This can be recognized as lobbying by businesses against the national legislature.

### 3.3. Deficiencies in the Foundation’s Operation Model

Compared to the previous state-management model, Laohegou Nature Reserve has adopted the foundation-management model, which can bring the advantages of specialization in ecological environmental protection and governance into full play, and it is more convenient for accessing and utilizing private capital. The model helps to solve the funding problem and meet the requirements of “ecology for the people, scientific utilization” in the guiding opinions, and has wide applicability. However, after operating for five years, the deficiencies of the Laohegou Foundation’s operation model gradually emerged. It includes the following three aspects.

Firstly, there is an imbalance between income and expenditure, which is self-limiting. The annual funding required in Laohegou Nature Reserve and the surrounding communities relies heavily on the Paradise feeding, and the annual funding is much higher than the income of the regurgitation-feeding projects. In 2018, Paradise contributed 5.25 million yuan to the project and operating costs, and also invested 370,000 yuan in the community’s projects of poverty alleviation by ecological means, directly purchased agricultural products and by-products of 1.5 million yuan, and provided an industrial start-up fund of 2 million yuan. In October 2015, Paradise established a wholly-owned subsidiary—Pingwu County Laohegou Baihuagu Honey Industry, LLC—with a registered subscribed capital of 3.75 million yuan. Although the company’s annual revenues are about 2.2 million yuan, it has borrowed money from Paradise for the company’s development for two years, with an accumulated 4.8 million yuan of outstanding debts by April 2017. In addition, the five villages of Minzhu, Shanhe, Fushou, Xinyi, and Guanba belong to the surrounding area of the Laohegou Nature Reserve, but the related industries are concentrated in Minzhu village, which is also the major recipient of financial funds from the Pingwu County government. Consequently, the unbalanced development of the community discourages the enthusiasm of other villages, which are also restricted by the protected areas in terms of production but have not been given equal development opportunities and have not been included in the green industrial chain of Laohegou. The spontaneous growth of the nature reserve, however, requires a broad mass base, and the goal of sustainable development cannot be achieved by unilateral investment alone.

Secondly, the limits of the organizational nature must be considered. The foundations are non-profit corporations engaged in the undertakings for the public interest. They can fulfill some duties of the government and be trusted by society. Therefore, their actions should strictly abide by the laws and regulations and avoid misappropriation in the name of public interest for personal gains or negatively affecting the social order. Both the Charity Law and the Regulations on the Administration of Foundations of China stipulate that the foundations can only carry out public interest operations specified in the regulations, with the only use of assets within the scope of the regulations, and cannot benefit specific natural persons, corporations, or organizations. Furthermore, Laohegou Nature Reserve Center, a subordinate unit established by Paradise, is a private non-enterprise unit. It is subject to the Interim Regulations on the Registration and Management of Private Non-Enterprise Units and can only engage in non-profit activities of social service, for there is no permission for non-profit organizations to directly engage in commercial operations. By establishing subordinate units as the “circumferential tactics”, commercial enterprises can evade the legislation. The foundations, non-enterprise units, and farmers’ professional cooperatives, however, have a different legal nature from the commercial companies, with different philanthropic particularism. In other words, this will cause an inevitable conflict of interest because of the different development targets and service objects. Thus, the approach of “separation first and then cooperation” will lead to an increase in the operating costs, and gradually diverging organizational goals, which will hinder the achievement of public interest targets [35].

Thirdly, flexible and rational use of natural resources in small private nature reserves is restricted. The Opinions on Comprehensively Promoting the Reform of Collective Forest Tenure System (hereinafter referred to as the “Opinions”), issued by the Chinese government, focus on easing the restrictions on the management right of forestland. The Opinions propose that non-commercial forests can also be rationally utilized, such as with the development of a planting and breeding industry and tourism in forestry land, without damage to the ecological functions. However, according to the requirements of the Regulations of Chinese Nature Reserves for core areas, buffer zones, and experimental areas, business activities can only be allowed in experimental areas. Apparently, that requirement seems to be too restrictive for the small privately protected lands, such as Laohegou Nature Reserve. Furthermore, the agreement signed between Paradise and the Pingwu County Government stipulates that non-commercial forests cannot be used for non-forest construction. In fact, by ensuring the main goals of nature reserves and the red line for resource consumption within a certain period, there is no need to overly restrict human activities which are reasonable. On the contrary, the proper activities help to mobilize the enthusiasm of social participation through their flexible operation so as to explore the mutual transformation of “lucid waters and lush mountains” and “invaluable assets” [36].

## 4. Advantages of Social Enterprises

### 4.1. Strong Self-Development Ability of Social Enterprises

If the foundation operation model in Laohegou Nature Reserve is the first generation of the organizational form of social participation, then the social enterprise is the second one. Social enterprises, which have overcome the deficiencies in the foundation engaging in the management of Laohegou, closely reflect the advantages of its participation in the management of protected areas, such as national parks. The dilemma in Laohegou reflects the voluntary failure of traditional non-profit organizations [37] for the lack of self-innovation and shortage of shareholders’ vitality [38]. On the one hand, traditional public interest undertakings rely so much on social donations that they are greatly affected by economic fluctuations and are not stable enough. On the other hand, non-profit organizations, such as foundations, cannot directly engage in for-profit activities, and even if the investment is allowed its amount is legally limited [39]. As a result, limited funding leads to philanthropic insufficiency [40]. In addition, owing to the two sides of the specialization of the foundation, it is likely to cause behavioral preferences and philanthropic paternalism, which leads to the loss of decision-making rights and the decline in the interest to participate for shareholders, such as community residents. Based on social innovation theory, social enterprises devote themselves to meeting the public’s demand for fair, efficient, and sustainable public services, and advocate the use of market and commercial means [41] by public interest organizations to innovate the approach. Therefore, the border between the traditional for-profit departments and non-profit departments gets a bit fuzzy, generating hybrid models. Social enterprises serve, first and foremost, a social mission. Furthermore, social enterprises accomplish a social mission through the use of sophisticated revenue-generating business models typically associated with traditional corporate activity [42]. Therefore, instead of relying on charitable donations and government subsidies, social enterprises can be self-reliant and determine their own direction of public welfare goals and choose a means of practice without the huge expense of relegating legislations or creating new operational models. The users pursuing public welfare goals can learn and adopt this model tool rapidly.

### 4.2. Solving Social Problems by Means of Business

Using commercial means to solve the insufficient momentum of traditional charity development is the biggest feature of social enterprises, that is, to participate in the market economy as an enterprise to achieve public interest goals [43]. It solves the problem of insufficient development strength in traditional public interest organizations. Social enterprises can maintain a single subject, withholding the public interest targets and reducing transaction costs. In addition, both the competitive market and the property rights system will continuously create external pressures, which will generate the internal motivation for social enterprises to improve efficiency and meet diversified needs [44]. Moreover, social enterprises can widely attract community residents to join the industrial chain by directly raising income, then prevent lazy behaviors, such as “free riding” and “relying on rescue”. Thanks to the organization of social enterprises, other entities, such as charitable organizations, including foundations and social organizations, public institutions, community shareholders, and even the general public, can join them in accordance with laws and articles as long as they agree with the public interest goals of social enterprises. The entities above can exercise the rights of decision-making and supervision through the investment of equity. Therefore, social enterprises can deliver long-term services of public interest in a more democratic and transparent environment.

### 4.3. The Accessibility of the Law Application

This merit of social enterprises plays a vital role in the reform of China’s institutions. The concept of social enterprise does not create a completely new type of organization but allows enterprises to operate in their existing form while the profits obtained are mainly used to solve problems of public interest [45]. Therefore, the legislative cost of social enterprises is low, which enables them to quickly participate in the construction of protected areas. China’s Civil Law divides social organizations into two categories, namely corporations, including for-profit corporations, non-profit corporations, and special corporations and, secondly, unincorporated organizations including sole proprietorships, partnerships, and professional service institutions without corporate capacity [46]. Considering that social enterprises use market operations as a means of subsistence and the requirements for their obligations, they should be included in the scope of for-profit corporations. China’s for-profit corporations include limited liability companies, joint stock limited companies, and other corporate corporations (one-person limited liability companies and wholly state-owned companies), and no matter how special the investor, all companies are consistent on the fundamental issue of obtaining profits and distributing them to investors. Furthermore, according to the provisions of the Law on Sino-Foreign Cooperative Enterprises and the Law on Foreign-Funded Enterprises, Sino-foreign cooperative enterprises and foreign-funded enterprises are for-profit organizations; hence, those who can obtain corporate capacity should also fall into the category of for-profit corporations. Therefore, as the most typical and significant type of body corporate in China, the company should become the main form of social enterprise. At present, many local governments in China have issued relevant policies and local regulations, such as the Measures for the Administration of Social Enterprises in Beijing (Trial Implementation) (Draft for Comments), the Measures for Certification of Social Enterprises in Beijing (2019) for Trial Implementation, the Measures for the Evaluation and Administration of Social Enterprises in Chengdu, and the Measures for the Registration and Administration of Social Enterprises in Neijiang. These policies or regulations fully prove that local governments recognize the advantages of social enterprises “giving consideration to both public and private interests”. Furthermore, the local governments express their confidence and enthusiasm to vigorously promote the development of social enterprises through legislation.

## 5. Key Collaboration Initiatives Activities

The characteristic of the market is the efficient pursuit of individual interests. Consequently, enterprises participating in public interest activities may be tempted to seek “self-interest”, which will lead to the depletion of social capital [47]. To ensure that social enterprises can create both social and economic value, it is necessary to establish an objective and specific organizational framework to win trust, and it also needs a mechanism to provide public interest to achieve the core spirit of social innovation. Only in this way can social enterprises exert a huge advantage over traditional public interest organizations, such as foundations, when participating in the construction of protected areas, such as national parks. The specific regulation methods are as follows.

### 5.1. Qualification Certification System of Social Enterprises

At present, social enterprises in China do not yet have a legal definition and status; even so, social enterprises are flourishing because they meet the needs of society. However, it will inevitably lead to fraud or false advertising without certain regulations. Social enterprise certification systems have been established in many countries. There are countries that take a parallel approach to government certification and social accreditation, such as the combination of British Community Interest Corporation (CIC) government certification [48] and Social Enterprise UK, the combination of Low-Profit Limited (L3C) Government Certification [49] and Benefit Corporation social certification [50]. Furthermore, certification systems in East Asia, such as in South Korea, Hong Kong, China, and Taiwan, China have also been formed [51]. Certification standards represent corresponding value systems, where social enterprises can form an alliance to overcome internal constraints and achieve resource sharing. Practices show that the construction of a qualification certification system is crucial to the healthy development of social enterprises. In order to set a benchmark for Chinese social enterprises, the CCF launched the “Social Enterprise Certification”. In the near future, social enterprises participating in the construction of protected areas will inevitably draw on the rich experience of the CCF to formulate social enterprise certification standards that not only boast the characteristics but also meet the construction goals of protected areas.

### 5.2. Asset Lock System for Social Enterprises

The vitality of social enterprises lies in sustainable development, while the original intention and core value of establishment are to “achieve public interest”. Therefore, it is necessary for social enterprises to establish a guaranteed mechanism for public interest [52]. With this guaranteed mechanism, whether they take the path of “non-profit organizations shifting from non-profit to for-profit” or “commercial organizations shifting from for-profit to public interest” [53], social enterprises can make a flexible plan according to actual market conditions and specific public interest goals. Moreover, when social enterprises provide services of public interest, they should not only take full advantage of their commercial profit capabilities but also prevent them from deviating from public interest to making profit. The “for-profit” of an enterprise means that it engages in business activities and distributes profits to investors [54]. The distribution of profits is one of the benchmarks for business operators. Even if the needs of shareholders are considered, it must be in line with the principle of maximizing the long-term interests of investors. In order to prevent social enterprises from distributing all profits to investors, it is necessary to set restrictions on the way they use their assets to ensure that the direction of public interest will not be changed [55]. Fortunately, the Company Law of China allows the articles of association of a company to show that no surplus profits shall be distributed to shareholders. Thanks to this, the company as a commercial entity with strong market vitality and capital cooperation manages to become the form carrier of social enterprises [56], and legally establishes the asset lock system.

The asset lock system has been widely used by social enterprises in European countries to lock fixed assets, revenue, and other profits. Specifically, it includes three restrictions on asset transfers, dividends (or performance interest) distributions, and surplus property distributions [57,58]. Firstly, the asset transfer restriction means that a social enterprise must include a clause in its articles of association, stating that when it transfers assets for the purpose of public interest, it must first obtain the permission of the administrative authority, and then they can only be transferred to other asset-locked entities with the same charter; that is, they can only be transferred to other non-profit organizations, such as social enterprises or foundations. Secondly, it imposes dividend distribution restrictions on social enterprises in the form of joint stock limited companies. Usually, only public interest subjects specified in the charter or the subjects approved by the administrative organs, a foundation with the same public interest goals, can be paid dividends. Therefore, when it comes to being distributed to shareholders, the articles of association must stipulate the upper limit of dividend distribution and cannot exceed the statutory upper limit. Thirdly, restrictions on the distribution of surplus property are enforced. When a social enterprise is dissolved, in order to ensure that the surplus property continues to be used for the undertakings of public interest and to prevent the enterprise from distributing assets to directors, shareholders, or interested parties in disguise, the shareholders can withdraw up to the nominal amount of the shares. Furthermore, the remaining assets or the proceeds from the sale of assets must be handed over to another asset-locked entity or administrative authority. A necessary method for regulating social enterprises is the asset locking system, which must be followed by social enterprises regardless of nationality, registered capital, and market size when they engage in the construction of protected areas in China. Thanks to these restrictions on assets, investors and other participants in social enterprises cannot be tempted by profits and remain true to their original aspirations to dutifully run their undertakings for the public interest.

### 5.3. Annual Reports of Public Interest

According to the Reform Plan of the Registered Capital Registration System approved by the State Council in February 2014, the former State Administration for Industry and Commerce of China (currently “State Administration for Market Regulation”) let the national enterprise annual inspection system be replaced by the enterprise annual report system in March 2014. Enterprises should report, in a timely fashion to the administrative organs for industry and commerce, the status of shareholders’ capital contributions, profits, and losses through the market entity credit information publicity system, and bear legal responsibility for the authenticity of the information. This reform not only conforms to the rules of the market economy and fully respects the right of enterprises to operate independently, but also strengthens credit supervision and coordinated supervision, and improves the efficiency of administrative supervision [59]. Social enterprises, which use the market for profit and adopt the corporate form, of course, are subject to deliver annual reports of public interest. Moreover, since social enterprises may enjoy preferential national policies in public interest activities, they should be required to make annual reports of public interest. The content of the reports should include public interest activities and financial information. Public interest activities should describe the company’s efforts in chasing its public interest goals in the past year, including the approach, progress, interaction with shareholders, and any other affairs that affect the goals; financial information should disclose the income and expenditure, profit distribution, and asset changes of social enterprises in terms of market operation, as well as the remuneration of directors and senior managers, and any financial or governance information between them and major shareholders holding more than 5% of shares that may have a substantial impact on the social credibility of enterprises. The annual reports of public interest should not only be uploaded to the information disclosure system and be subject to random inspection by the administrative authority, but also provided to shareholders at the annual shareholders’ meeting and disclosed on the company’s website for public inspection. With the submittal of the annual reports of public interest, costs are saved for investors and consumers, and it further prevents social enterprises from pursuing private interests, with information asymmetry mitigated while the outside supervision is strengthened. In addition, social enterprises should be forced to put shareholders onto the catalog of decision-makers by the means of holding symposiums, opening an account on the popular microblogging service Sina Weibo, and other ways. Alternatively, social enterprises must obtain the consent of shareholders before making specific decisions, which should be stipulated in the constitution and be included in the annual reports of public interest. The above measures can encourage social enterprises to strengthen self-discipline, reduce service costs, and improve service quality [59].

### 5.4. Triple Supervision and Accountability Mechanism

First of all, there is no doubt that social enterprises, with the structure and function of the commercial entity, can use the internal institutions stipulated in the Commercial Law to supervise themselves [60]. As mentioned above, companies serve as the best carriers for social enterprises at both the expected and actual levels, whose advantages include direct access to the company’s mature supervisory mechanisms. This can be reflected in the following aspects. The general meeting of shareholders, the power organ of the company, exercises the legal rights to elect or dismiss directors, amend the articles of association of the company, and decide on major transactions. Furthermore, it can also extensively monitor the behavior of operators. Additionally, the Company Law imposes a fiduciary duty on directors at a company, which can also apply to social enterprises [61]. In for-profit businesses, fiduciary duty means that directors and senior management are trusted by funders and should be diligent in for-profit undertakings. However, in social enterprises, the connotation of fiduciary duty has changed from “profit-oriented” to “public interest-oriented”, with profit-making being a means to achieve this goal [62]. In other words, directors of social enterprises have an obligation to use the company’s assets for the defined public interest, which is the starting point and end point of their decisions. In addition, the supervisory and financial accounting systems stipulated in the Company Law should also be fully introduced in social enterprises, for only the integrated application can guarantee the effectiveness of internal control.

Social enterprises should be recognized as general market entities, which are subject to the supervision of administrative authorities in charge of industry and commerce, taxation, and so on. Besides this, there is also external supervision in the specific area of public welfare. When social enterprises participate in the construction of protected areas, there is no doubt that they should fulfill their legal obligations and quickly reach the defined targets, whether based on authority or contract, in the supervision of departments of ecology and environment, and protected land management bodies. However, the supervision of power should not be too strict; hence, it is not appropriate to use the standards of traditional public interest organizations, such as foundations. On the contrary, the conditions for exercising supervision shall be clearly defined, and normal business activities should not be interfered with. Finally, democratic supervision is an important protective system of internal supervision and administrative power supervision. Obtaining a market reputation and social trust can benefit the growth of social enterprises. Thanks to democratic supervision with a powerful deterrent, it can not only prevent shareholders from neglecting to exercise their supervisory power and even collective deception, but also enhance the efficiency of supervision in administrative organs, and can prevent rent-seeking by power. The protected areas must establish an effective democratic supervision mechanism and accept more strict media supervision, with the measures of reporting reward and return visits. When it comes to empowering citizens’ rights to information and supervision, the relevant law should not only provide for “information disclosure”, which is a single, low-level method. Instead, government departments should be required to respect the right of the news media to interview, and not refuse it without justifiable reasons. For the illegal problems of social enterprises exposed through “We Media”, it is necessary to deal with them according to the procedures and disclose the result shortly after the exposure, and to give administrative rewards to the citizens who report the truth.

### 5.5. Third-Party Testing Mechanism

Third-party testing, also known as impartial testing, refers to the evaluation of goods or services by third-party institutions other than the government and enterprises, as non-parties according to laws, contracts, or relevant standards. The third-party institution can be entrusted by both parties of the transaction, or it can be independent of both parties, and it can also be entrusted by the government, social organizations, or the public [63]. In the 15th century, third-party testing appeared in European and American countries. With the development of the market economy and the improvement of science and technology, it had become so mature and widespread by the middle of the 19th century that many testing institutions with high reputations and great industrial influence had been produced. Affected by flexible factors, the third-party testing mechanism in China developed late. A large number of foreign testing institutions did not flood China and rapidly occupy the market until twenty or thirty years ago. On the one hand, with export trade booming, third-party testing should be carried out according to the requirements of buyers. On the other hand, consumers do not trust the detection of government and industry associations under the high-power administrative power. Third-party testing has become a mature industry, providing the public with evaluation results other than from the government, which can effectively regulate the production and operation of enterprises. The third-party testing industry should be actively developed so as to drive the growth of social enterprises.

The third-party testing of social enterprises includes two aspects, as follows: one is the products or services marketed on the market, and one is public interest activities and social influence. The third-party inspection must meet the following conditions: (1) Comprehensiveness. It shall comprehensively assess the public interest activities of social enterprises, including the impact of the project on shareholders; (2) Independence. The evaluated social enterprise has no control or interest relationship with the testing agency, and testing results from the third party entrusted by the social enterprise have no credibility beyond advertising effectiveness; (3) Credit. Third-party testing institutions must be equipped with professional qualifications and testing experience; (4) Transparency. No one is allowed to hide the test results. Therefore, information, such as evaluation indicators, weighted components, and standard changes, should be disclosed to the public for public supervision.

## 6. Conclusions

The national parks are devoted to promoting the co-prosperity of human beings and the ecological environment. Traditional public interest organizations, such as foundations, are involved in the management of the park, which keeps up with the times and meets the needs of society. The foundation operation model of Laohegou Nature Reserve is the initial fruit of the joint development of between environmental public interest by the government and social forces. The model has shown advantages and a variety of application areas. Social enterprises upgrade the public participation model, with its biggest feature being in “giving consideration to both public and private interests”, to mobilize social capital and strength to join public interest projects under minimal government spending. Moreover, it maintains the operation of public interest projects with income. In conclusion, it effectively lightens the financial burden on the government, which is of great significance in the post-COVID-19 epidemic era when government input is needed in all aspects. Furthermore, in solving the targeted problems, social enterprises are not only market entities with self-reliance, but also a public interest force with trust. By adopting flexible management and broad business ideas as market entities, social enterprises can perfectly match the different types and sizes of protected areas to provide innovative, diversified, and characteristic services. Under the United Nations Sustainable Development Policy and the guidance of “service for the people” and “people-oriented” spirits, the government should actively improve undertakings for public interest in environmental protection, and create a policy environment with clear objectives, self-consistence in operating logic, and coordinated cooperation among various entities. The participation of social enterprises in the institutional construction of protected areas is an experiment in public participation in the post-COVID-19 epidemic era, which can promote the establishment of a system of protected areas with characteristics, and provide institutional innovation and practical demonstrations for the world.

## Figures and Tables

**Figure 1 ijerph-19-16278-f001:**
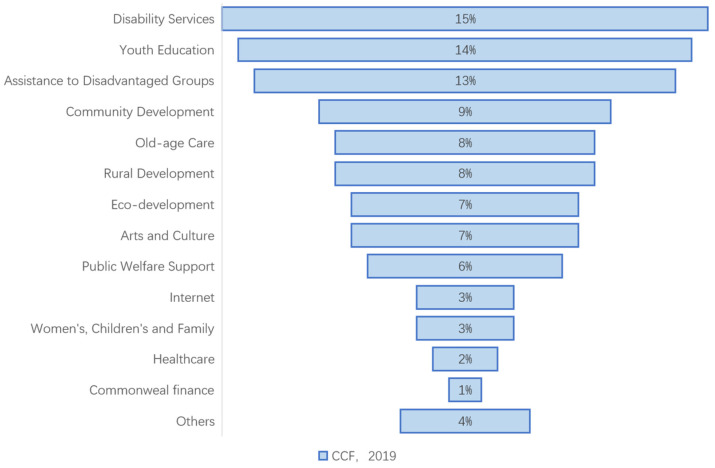
Field distribution of social enterprises.

**Figure 2 ijerph-19-16278-f002:**
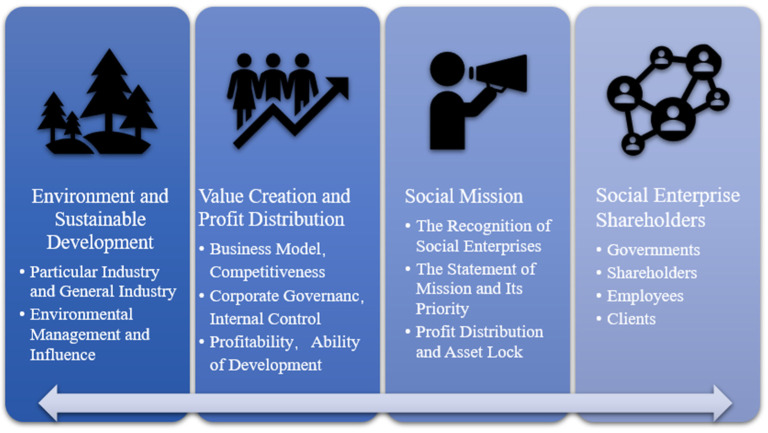
Four-dimensional indicators of social enterprises.

**Figure 3 ijerph-19-16278-f003:**
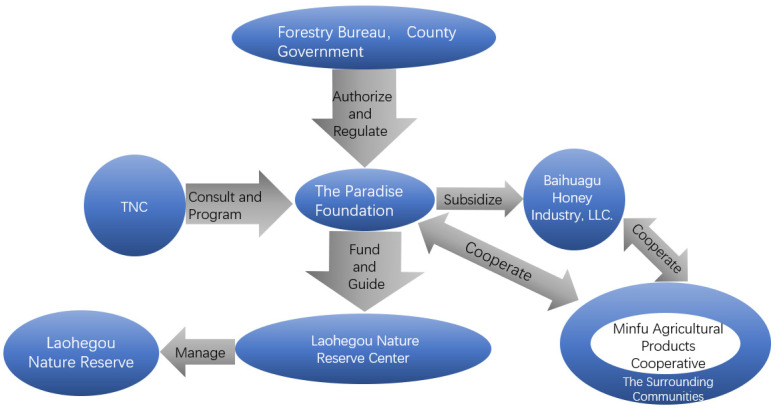
Organizational structure of the Laohegou Nature Reserve.

**Table 1 ijerph-19-16278-t001:** Public service delivery modes under different theoretical frameworks.

Theory	Public Service Delivery Modes	Disadvantages	Aftermath
Omnipotent government	Government	Excessive pressure on government finance; loss of Flexibility	Government failure
New public management	For-profit enterprises	Unfair competition; overpriced public services	Market failure
New public service	Non-profit organizations	Relies heavily on government finance; loss of autonomy after market financing	Voluntary failure

**Table 2 ijerph-19-16278-t002:** The Development Stages of Laohegou Nature Reserve.

Stage	Ownership	Operators	Items
Forest farm period	State-owned forest farm	Forestry Bureau, County Government	Logging in natural forests, planting economic forests
Stagnation period	Stated-owned forest farm	Forestry Bureau, County Government	Protecting natural forests, planting man-made forests
Nature reserve construction period	State-owned forests and collective-owned forests	Forestry Bureau, County Government; rural collective	Ecological protection of non-commercial forests, tourism, planting, and logging of commercial forests
Private capital period	State-owned forests and collective-owned forests	Sichuan Paradise International Foundation	Ecological conservation, tourism, characteristic agriculture, and others

## Data Availability

The data presented in this article is available in the public domain in the following locations: Google Scholar, Web of Science, Medline, and PubMed.

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
