# Peer review of "Innovation of the Third Sector to Improve Nature Reserve Management in the Post-COVID-19 Epidemic Era"

_ijerph, 2022, doi:10.3390/ijerph192316278_

Round 1

Reviewer 1 Report

By analyzing the structure and function of social enterprises, this article attempts to prove that social enterprises do help to solve "government failure" and "market failure" problems in the management of nature reserves. Obviously, the structure and function of social enterprises are designed to deliver public goods commercially (theoretical research is abundant). However, it is not clear how social enterprise operates in actual cases. This article fills the research gap by taking the management practice of Laohegou Nature Reserve as a case study. 

General recommendations: 

  1. Laohegou case explains that the three-dimensional pattern is beneficial in general. It is better to analyze the Laohegou Nature reserve case deeply. For example, add data to show and explain whether the management efficiency has improved with the help of social enterprise, in which aspect (e.g., the protection of the landscape, soil, biodiversity, traditional knowledge, culture, and the creation of jobs) and explain why (attributable to the particular structure or function of social enterprises?). 
  2. Social enterprises can be utilized to provide a diverse range of public goods. The novelty of this research lies in that it stresses the nature reserve. This aspect can be improved by explaining the specialty of the nature reserve in the introduction part or case study part (e.g., the requirement of the management of nature reserves is different from that of other kinds of public services). 
  3. If this article is to stress the case study, which is the critical innovation part, then highlight it in the title.
  4. The following two articles are good examples of how to use case studies to show how theories operate in real cases (not about social enterprises but we can learn their logic): Product qualification as a means of identifying sustainability pathway for place-based agri-food systems: the case of the GI Corsican grapefruit (France) (Maorgane Millet and others 2020); The potential of Geographical Indications (GI) to enhance sustainable development goals (SDGs) in Japan: overview and insights from Japan GI Mishima potato (Junko Kimura, and Cyrille Rigolot 2021).

Specific recommendations:

1) Concerning the title, please clarify: to improve nature reserve management, do you mean to improve management efficiency, performance, or other concerns?

2)   Concerning omitted citations: Please disclose the source of the following parts: CCF defines social enterprises as "…" (Page 5), (in a book? Or on the Web?); the data about Laohegou (pages 8-9) and the organizational structure of the Laohegou Nature Reserve (Page 10-12); the "opinions" proposes that…(Page 14, which article?); "China's civil law divides…." (Page 15, which article?); Social enterprise certification systems have been established in 22 countries (page 16, where the data comes from?).

3)  Concerning the subtitle or the first sentence of one paragraph, sometimes, the sub-title or the first sentence means one thing, but the content means another. For example, "Thirdly, unreasonable use of natural resources under the conflicts of laws" (p. 14). Actually, this sentence would like to argue that the law restricts the use of resources even when it is not harmful. Please check the suitability of titles 3.3 (the imbalance between income and expenditure does not belong to the deficiencies in the operational model, it is the result of the deficiencies) and 4.1, or you may save the current titles and re-organize the contents (arguments).

4) This article argues that "the empirical research indicates that social enterprise needs to be supplemented with the maintenance mechanism of the bidirectional targets…" in the abstract. If the empirical research refers to the Laohegou case, then please combine the case with the suggestions concerning the maintenance system (Chapter 5 Key collaboration initiatives activities, e.g., clarify the connection between the strength or weakness of the asset lock system, the annual report, and the supervision mechanisms of Laohegou nature reserve organization and the management performance).

Author Response

Dear reviewer,

        thank you very much for your patient review, your recognition of the article and pertinent suggestions are greatly appreciated.

     In response to the General recommendations, thank you very much for providing two articles for the authors to study, which will help us a lot in our subsequent research and thesis enhancement! Regarding the structure of this article, the authors' intention is to introduce social enterprises as a new third sector first, then show the advantages of social enterprises through the comparison with traditional foundation models in practice, and finally propose how to make social enterprises serve the management of nature reserves better, and gives specific measures. The authors have revised the title of the third part to "Advantages of Social Enterprises - Through Comparison with Foundation Model in Practice".

        In response to specific recommendations,

    1) Social enterprises provide an alternative operating model for nature reserves, and the effects are multifaceted, including improved management efficiency, balanced revenues and expenditures, and broader community participation. Therefore, the authors think it is appropriate to maintain the current title.

       2) Sources have been added in the citations in the footnotes. In addition, for the data on page 16, it comes from a presentation at an academic conference attended by the authors. The authors have replaced "22" with "many", because the reporter's paper was not published and no further citation details can be given. Thank you very much for your correction, and we hope this change will not affect the reading effect.

    3) The first sentence has been modified in line 457. For section 3.3,the foundation is not suited to assume a leading role in the nature reserves due to its institutional structure and organizational mission, and the imbalance between income and expenditure is indeed a consequence of its natural structural shortcomings. The authors think it is more in line with the meaning of the content without modification. For section 4.1, we add how the strong capacity of social enterprises to develop themselves is evident, as detailed in the Annex.

    4) The article's approach is to introduce social enterprises as a new third sector. The first nature reserve managed by the third sector in China is Laohegou, which uses the traditional foundation model. The problems revealed by Laohegou are exactly what social enterprises can remedy. Since social enterprises are just starting out in China, there is no precedent in nature reserves’ management. The fifth part is a comprehensive theoretical elaboration about how to use social enterprises model, combined with the characteristics of nature reserve management and the problems exposed by Laohegou. The authors will strengthen the simulation control in subsequent studies, but for this paper, a completely new article structure may be required, the authors decided not to make changes on balance.

        The formatting and duplication issues have been fixed.

        We thank you again for your pertinent suggestions, which have benefited us greatly.

                                                                         2022.12.01

Reviewer 2 Report

The article is well structured, setting the context in the "post-epidemic period" and limiting the scope to "protected area construction" to introduce the concept of "social enterprise". The argument of the article is complete, especially it presents the existing problems of traditional model and introduces the new-type sector: social enterprise. Moreover, the paper gives proposals for further regulation.

However, the quality of the manuscript shows that some aspects should be improved. Thus, my recommendation for the editor is a minor revision. To improve the quality of the paper, I provide some comments:

1. The concept of "social enterprise" is not clearly defined throughout the text. As there is no corresponding law in China that defines social enterprises, this concept in the article is somewhat vague and inconsistent. The authors should give a clear definition.

2. The connection between Part III and Part IV is incomplete. The third part shows the case of Laohegou Nature Reserve, and the fourth part suddenly goes to show the advantages of social enterprises. The logic here should be as follows: First, the model of Laohegou Nature Reserve, as the first generation of the organization form of social participation, has some drawbacks on management. Second, these deficiencies could be overcome by the new organization form that the second generation of social enterprise is introduced. Finally, the advantages of social enterprise could be developed to discuss. These two parts lack a corresponding transition.

3. Part 4.1 of the article involves "Strong Self-Development Ability of Social Enterprises", but the authors devote a lot of space to writing about the disadvantages of the traditional model, and only mention the self-development ability of social enterprises at the end of the paragraph. In other words, the association between the content and the title needs to be strengthened.

4. There are some typos and formatting issues, for example repetitions in lines 106-113 of the paper regarding “entrepreneurial government”.

Author Response

Dear Reviewers,

      thank you very much for your patient review, your recognition of the article and pertinent suggestions are greatly appreciated.

       In response to your recommendations, the authors have made the following changes:

  1. In section 2.2, the authors add our own definition of social enterprise (P.5 line 156-161).
  2. Your suggestion that the article should strengthen the logical linkage is very valuable. The authors have revised the title of the third part to "Advantages of Social Enterprises - Through Comparison with Foundation Model in Practice". In section 3.3. Deficiencies in the Foundation's Operation Model, we point out the shortcomings of the first-generation "foundation" model for Laohegou Nature Reserve (lines 407-412 of P.13), and in section 4.1. Strong Self-Development Ability of Social Enterprises, we highlight the characteristics and advantages of social enterprises (lines 495-502 of P.15).
  3. Regarding the structure of thisarticle, the authors' intention is to introduce social enterprises as a new third sector first, then show the advantages of social enterprises through the comparison with traditional foundation models in practice, and finally propose how to make social enterprises serve the management of nature reserves better, and gives specific measures.For section 4.1,we add how the strong capacity of social enterprises to develop themselves is evident, as detailed in the Annex.
  4. The formatting and duplication issues have been fixed, thank you for your careful review.

      We thank you again for your pertinent suggestions, which have benefited us greatly.

                                                                                    2022.12.01

Reviewer 3 Report

I found the contents of this paper to be very interesting, but I had to work hard to understand it.  Partly this is because I have a biology/environment background, not a law and economics background. Also, I am not familiar with the legal and land management systems used in China. But these are also the reasons that the paper is interesting. I see the importance of the study to nature reserve management. 

Based on my comments in the attached file, you might consider adding some description about your own methods of compiling the information; you might consider splitting Section 3 into a section more specifically about Methods, then a new section (currently 3.2 and 3.3) or Results; and consider whether you can make more cross-references between current sections 3.2 and 3.3 with the material in sections 4 and 5.

But overall, I like this paper, and it contains a lot of important information that will be of interest to an international audience.

Author Response

Dear Reviewer,

        thank you very much for your patient review, your recognition of the article and pertinent suggestions are greatly appreciated.

     First of all, we apologize that the authors' disciplinary backgrounds and language styles may cause some obstacles to your reading. Your suggestions will help us a lot in our subsequent research and thesis enhancement. Regarding the structure of this article, the authors' intention is to introduce social enterprises as a new third sector first, then show the advantages of social enterprises through the comparison with traditional foundation models in practice, and finally propose how to make social enterprises serve the management of nature reserves better, and give specific measures.

        In response to your review comments, the authors have made the following changes:

        1) Your suggestions remind us that the article should strengthen the logical linkage In Sector 3-Sector 4. The authors have revised the title of the third part to "Advantages of Social Enterprises - Through Comparison with Foundation Model in Practice". In section 3.3. Deficiencies in the Foundation's Operation Model, we point out the shortcomings of the first-generation "foundation" model for Laohegou Nature Reserve (lines 407-412 of P.13), the foundation is not suited to assume a leading role in the nature reserves due to its institutional structure and organizational mission, and the imbalance between income and expenditure is indeed a consequence of its natural structural shortcomings. In section 4.1. Strong Self-Development Ability of Social Enterprises, we highlight the characteristics and advantages of social enterprises, we add how the strong capacity of social enterprises to develop themselves is evident (lines 495-502 of P.15), as detailed in the Annex. And the first sentence has been modified in line 457.

       2) Sources have been added in the citations in the footnotes. In addition, for the data on page 17(line. 566), it comes from a presentation at an academic conference attended by the authors. The authors have replaced "22" with "many", because the reporter's paper was not published and no further citation details can be given, and we hope this change will not affect the reading effect.

     3)The article's approach is to introduce social enterprises as a new third sector. The first nature reserve managed by the third sector in China is Laohegou, which uses the traditional foundation model. The problems revealed by Laohegou are exactly what social enterprises can remedy. Since social enterprises are just starting out in China, there is no precedent in nature reserves’ management. The fifth part is a comprehensive theoretical elaboration about how to use social enterprises model, combined with the characteristics of nature reserve management and the problems exposed by Laohegou. The authors will strengthen the simulation control in subsequent studies, but for this paper, a completely new article structure may be required, the authors decided not to make changes on balance.

        4) In section 2.2, the authors add our own definition of social enterprise (P.5 line 156-161).

        5) The formatting and duplication issues have been fixed.

       We thank you again for your pertinent suggestions, which have benefited us greatly.

                                                                                      2022.12.01